# Bioprinting on 3D Printed Titanium Scaffolds for Periodontal Ligament Regeneration

**DOI:** 10.3390/cells10061337

**Published:** 2021-05-28

**Authors:** Ui-Lyong Lee, Seokhwan Yun, Hua-Lian Cao, Geunseon Ahn, Jin-Hyung Shim, Su-Heon Woo, Pill-Hoon Choung

**Affiliations:** 1Department of Oral & Maxillofacial Surgery, Dental Research Institute, School of Dentistry, Seoul National University, Seoul 03080, Korea; davidjoy76@gmail.com (U.-L.L.); wild525@naver.com (H.-L.C.); 2Department of Oral & Maxillofacial Surgery, Chung-Ang University Hospital, Seoul 06973, Korea; 3Department of Mechanical Engineering, Korea Polytechnic University, Siheung-si 15073, Gyoenggi, Korea; yuntobi@kpu.ac.kr (S.Y.); happyshim@kpu.ac.kr (J.-H.S.); 4Research Institute, T&R Biofab Co., Ltd., Siheung-si 15073, Gyoenggi, Korea; gsahn84@gmail.com; 5Research Institute, Sphebio Co., Ltd., Pohang-si 37666, Gyeongsanbuk, Korea; 6R & D Center, Medyssey Co., Ltd., Jechon-si 27159, Gyeongsanbuk, Korea; suheon.woo@medyssey.com

**Keywords:** 3D printing, dental implant, periodontal ligament, stem cell, scaffold

## Abstract

The three-dimensional (3D) cell-printing technique has been identified as a new biofabrication platform because of its ability to locate living cells in pre-defined spatial locations with scaffolds and various growth factors. Osseointegrated dental implants have been regarded as very reliable and have long-term reliability. However, host defense mechanisms against infections and micro-movements have been known to be impaired around a dental implant because of the lack of a periodontal ligament. In this study, we fabricated a hybrid artificial organ with a periodontal ligament on the surface of titanium using 3D printing technology. CEMP-1, a known cementogenic factor, was enhanced in vitro. In animal experiments, when the hybrid artificial organ was transplanted to the calvarial defect model, it was observed that the amount of connective tissue increased. 3D-printed hybrid artificial organs can be used with dental implants, establishing physiological tooth functions, including the ability to react to mechanical stimuli and the ability to resist infections.

## 1. Introduction

Clinicians and researchers are constantly searching for ways to improve the longevity of dental implants, especially when the alveolar bone is highly resorbed and the retention and stability of the existing root implants are low. If there is not enough bone, patient-specific implants may be a good choice for the installation of dental implants. Additive manufacturing enables the fabrication of custom implants at microscale resolution, presenting as a potential technology for the manufacture of dental implants [1]. 

The development of CAD/CAM technology and electron beam melting (EBM) with adequate strength and biocompatibility provide the foundation for the application of patient-specific titanium 3D-printed implants during maxillofacial bone reconstruction [2,3,4]. In fact, a multicenter study using patient-specific CAD/CAM reconstruction plates for mandibular reconstruction has been performed with good results [5]. Another study reported the successful reconstruction of maxillomandibular defects using 3D-printed titanium mesh [6].

This additive manufacturing method has benefits over the existing titanium computerized control (CNC) milling method, as the internal configuration of the implant can be designed as desired. As the surface of the implant can be fabricated with a mesh structure, the increased porosity of the implant surface improves cell proliferation and mesenchymal stem cell differentiation, resulting in the promotion of osseointegration [2,7]. Two types of osseointegrated dental implant models, such as endosteal implants and subperiosteal implants, can be fabricated by the titanium 3D printing technique [1].

Osseointegrated dental implants are very reliable and have long-term reliability. However, host defense mechanisms against infection have been known to be impaired around dental implants because of the lack of a periodontal ligament (PDL) [8,9].

To solve this problem, a 3D-printed hybrid artificial organ was fabricated by combining EBM technology and bioprinting of the PDL. PDL protects against infection and bone resorption associated with mechanical stress, such as traumatic occlusal force and orthodontic tooth movement [8,9].

Currently, 3D cell-printing techniques have been utilized as new biofabrication platforms because of their ability to precisely locate living cells in pre-defined spatial locations with scaffolds and various growth factors [10]. 3D bioprinting is the utilization of 3D printing and 3D printing–like techniques to combine cells, growth factors, and biomaterials to fabricate biomedical parts that maximally imitate natural tissue characteristics [10,11,12,13,14,15,16]. A 3D bioprinted skin patch mixed with adipose-derived stem cells enhanced wound closure, re-epithelization, and neovascularization [10]. Implanted, bioprinted, neural stem cells in adult zebrafish accelerated repair of traumatic brain injury and restoration of function [11]. Implanted, bioprinted cell types including human amniotic-derived stem cells, rabbit ear chondrocytes, and rabbit myoblasts produced newly formed vascularized bone tissue, cartilage, and muscle with physiologically relevant mechanical properties [12]. The PDL is the fibrous connective tissue covering the root of the tooth, and the alveolar bone that forms the socket wall [13]. PDL protects against infections and bone resorption associated with mechanical stress [14,15]. The presence of a PDL can permit a dynamic role even in a functionally ankyloses osseointegrated implant [16]. PDL-derived cells have stem cell-like characteristics and are regarded as versatile sources for periodontal reconstruction [13]. In this study, we tried PDL regeneration on 3D-printed titanium structure’s surface using the human periodontal ligament stem cells (hPDLSCs) bioprinting technique for fabricating 3D-printed hybrid artificial organ.

## 2. Materials and Methods

### 2.1. Primary Cell Culture from Extracted Human Third Molar

Non-decayed human 3rd molars that had been impacted in the mandible were extracted from 5 adults (18–28 years of age) with informed consent at the Seoul National University Dental Hospital, Seoul, South Korea. The experimental protocol was approved by the Institutional Review Board of the hospital (IRB No. 05004). The PDL was gently separated from the root of the extracted 3rd molars, and the separated tissues were digested in a solution of 3 mg/mL collagenase type I (Worthington Biochem, Freehold, NJ, USA) and 4 mg/mL dispase (Boehringer, Mannheim, Germany) for 1 h at 37 °C. Single-cell suspensions were collected by passing the cells through a 40 mm strainer (Falcon BD Labware, Franklin Lakes, NJ, USA) and were cultured in the alpha-modification of Eagle’s medium (α-MEM; Gibco BRL, Grand Island, NY, USA) supplemented with 10% fetal bovine serum (Gibco BRL), 100 mM ascorbic acid 2-phosphate (Sigma-Aldrich, St. Louis, MO, USA), 2 mM glutamine, 100 U/mL penicillin, and 100 mg/mL streptomycin (Biofluids, Rockville, MD, USA). The medium was changed after the first 24 h and then every 3 d. A total of 3 colonies of hPDLSCs were randomly picked, and the cellular pool of these colonies was used for in vitro proliferation, differentiation studies, and animal experiments. All primary cells used in this study were in cell passage stages 2 or 3. 

### 2.2. Flow Cytometric Analysis of the hPDLSCs

To characterize the immunophenotype of the hPDLSCs, the expression of mesenchymal stem cell-associated surface markers at passage 3 were analyzed by flow cytometry, as previously reported [17]. hPDLSCs in their 3rd passage (1.0 × 10^6^ cells) were fixed with 3.7% paraformaldehyde from 95% paraformaldehyde powder (Sigma-Aldrich) diluted in phosphate-buffered saline (PBS) (3.7 g/100 mL) for 10 min and re-suspended in PBS containing 1% bovine serum albumin (BSA) (ICN Biomedicals) for 30 min for blocking nonspecific antibody-binding sites. hPDLSCs were then incubated with specific antibodies against CD34, CD13, CD90, or CD146 at 4 °C for 1 h, and then incubated with fluorescent secondary antibodies at room temperature for 1 h. All used antibodies were purchased from BD Biosciences. The percentages of CD13-positive, CD90-positive, CD146-positive, and CD34-negative cells were measured using a FACS Calibur flow cytometer (Becton Dickinson Immunocytometry Systems). The results were analyzed by CellQuest Pro software (Becton Dickinson).

### 2.3. Osteogenic, Chondrogenic, and Adipogenic Differentiation of the hPDLSCs

To promote osteogenic, chondrogenic, and adipogenic differentiation, hPDLSCs were cultured in StemPro Osteogenic, StemPro Chondrogenic, and StemPro Adipogenic differentiation medium (Gibco BRL), respectively, with the appropriate supplements as previously reported [17]. At 21 days, the cells with post-osteogenic, post-chondrogenic, and post-adipogenic induction were stained with 2% Alizarin Red S stain at pH 4.2 (Sigma-Aldrich), 1% Alcian Blue (Sigma-Aldrich), and 0.3% Oil Red O dye (Sigma-Aldrich) to detect proteoglycans, Nissl bodies, and fat vacuoles as indicators of osteogenic, chondrogenic, and adipogenic differentiation, respectively. Stained cells were observed and those were visualized under an inverted light microscope (Olympus U-SPT; Olympus).

### 2.4. 3D Printing

Two CAD/CAM programs (3-Matic/MAGICS, Materialize, Leuven, Belgium) to design a 3D-printed implant for hPDLSCs printing were utilized. The porous structures were based on dode-thin unit cells in the MAGICS program with the following design (nominal) dimensions: strut size = 120 μm, pore size = 500 μm and porosity = 88%. Disk-shaped samples with size (Ø8 mm × H2 mm, Ø18 mm × H2 mm) were designed. The designed STL file was programmed to a 3D printer with an EBM method of metal additive manufacturing (Arcam A1, Arcam, Gothenburg, Sweden), and samples were printed using Ti-6Al-4 V-ELI medical-grade powder (Arcam A1), as previously described. 

### 2.5. Bioprinting

The experimental groups used in this study were only 5% (*w*/*v*) collagen (Union Pharma, Seongnam, Korea) and 5% collagen mixed with 10 ng/mL FGF-2 (FGF-2, ProSpec, Rehovot, Israel). For bioprinting of a PDL layer on titanium scaffolds, hPDLSCs were added to the bioink mixture (1 × 10^7^ cells/mL). Titanium scaffolds fabricated via 3D printing were prepared for bioprinting of the PDL layer. In this study, experimental groups were classified according to cell seeding and cell printing methods. In the cell seeding group, collagen or collagen/FGF-2 inks without cells were printed on a titanium scaffold surface and stored for 30 min at 37 °C for gelation. After that, hPDLSCs were seeded on the printed ink layer group 1 (G1): titanium scaffold/collagen/cell seeding, group 2 (G2): titanium scaffold/collagen+FGF-2/cell seeding). In the cell printing group, the bioinks with cells group 3 (G3): titanium scaffold/collagen/cell printing, group 4 (G4): titanium scaffold/collagen+FGF-2/cell printing were printed on a titanium scaffold surface and stored for 40 min at 37 °C for gelation (Figure 1). For precise cell printing experiments, a bioprinter was used (3DX Printer, T & R Biofab Co., Ltd., Siheung, Korea) (Figure 2). Bioprinting was performed using a disposable nozzle with an inner diameter of 400 μm. All samples were cultured under the same conditions as hPDLSCs. The viability of printed hPDLSCs was evaluated using a live/dead cell assay kit (Lonza, Walersville, MD, USA). The proliferation of printed hPDLSCs was evaluated using a CCK-8 kit (Dojindo, Gumamoto, Japan).

### 2.6. Scanning Electron Microscopy

Four groups of samples were fixed with modified Karnovsky’s fixative for 2 h. The samples were washed thrice with PBS buffer for 15 min and fixed with 1% Osmium tetroxide (EMS). The samples were then washed with distilled water and dehydrated with graded concentrations (70, 80, 90, 95, and 100% *v*/*v*) of ethanol. The samples were then treated with hexamethyl disilazane (HMDS) for 20 min. Finally, the samples were coated with Pt prior to cell shape observation using a field-emission scanning electron microscope (FE-SEM; Hitachi S-4700) using an acceleration voltage of 15 kV at 3 different magnifications: ×10,000, ×1000, and ×50. The 4 groups of samples were incubated for up to 21 d at 37 °C in 5% CO_2_ to determine how long the gelled collagen printed on the scaffold retained its shape and pictures were taken every day. 

### 2.7. LIVE/DEAD Assay and Quantitative Polymerase Chain Reaction

A LIVE/DEAD Viability/Cytotoxicity Kit (Molecular Probes, Eugene, OR, USA) of the cell-printed structures at day 1 was used for checking that the cells had been printed properly. The sample was observed with a fluorescence microscope (Axiovert 200, Zeiss, Jena, Germany).

To evaluate PDL gene expression levels, 4 groups of samples were cultured for 7 d without any induction. Total RNA was prepared using a RNeasy Mini Kit (Qiagen, Germantown, MD, USA) according to the manufacturer’s instructions. cDNA was synthesized from 1 μg of total RNA using reverse transcriptase (Superscript II Preamplification System; Invitrogen, Waltham, MA, USA). Oligonucleotide primers for the amplification of human ALP, CEMP1, and Col1 mRNA were designed. Real-time PCR was performed with SYBR Green PCR Master Mix (ABI Prism 7500 sequence detection system, Applied Biosystems, Waltham, MA, USA). The reaction conditions were: 40 cycles of 15 s of denaturation at 95 °C and 1 min of amplification at 60 °C. All reactions were run in triplicate and were normalized to the reference gene (GAPDH). 

### 2.8. PDL Regeneration In Vivo

Athymic rats (Hsd:RH-Foxn1Rnu, 10 males, 9 weeks old, Envigo, NJ, USA) were used for the animal experiments involving the transplantation of the 3D-printed titanium scaffolds with seeded or cell-printed hPDLSCs into a calvarial bone defect. The study protocol was designed according to the ARRIVE guideline and was approved by the Ethics Committee on Animal Experimentation of Chung-Ang University (2018-00091) on 03 August 2018. All animals were housed in the same specific pathogen free (SPF) facility under a 12 h light/dark cycle with ad libitum feeding. No more than 3 animals were placed in a single cage. A total of 8 animals per group were carried out randomized in 4 groups as in the group in the in vitro experiment. Two 3D-printed titanium scaffolds with seeded or cell-printed hPDLSCs were transplanted into 8 nude rats each.

The sample size per group was minimized as recommended by the Ethics Committee on Animal Experimentation of Chung-Ang University. The rats were anesthetized with alfaxanolone (Alfaxan^®^, Jurox, Australia) (3 mg/kg) and xylazine hydrochloride (Rompun^®^, Bayer Korea, Korea) (10 mL/kg) IP. To access the cranial vault of the rat, a 4 cm incision was made in the skin with a number 15 blade in the middle region of the skull. The skin, facial tissue, and periosteum were dissected to gain enough access to the calvaria. Using an 8 mm trephine bur, 2 circular-shaped bone ditches were made with a diameter of 8 mm that did not penetrate into the dura. Using a round bur, a bony pit with a depth of 1.5 mm inside the circle was made on the calvaria. Four groups of samples after 2 d of incubation were transplanted with the cell layer in contact with the calvarial bone. After transplantation, the surgical wound was sutured with 4–0 nylon sutures. Six weeks after transplantation, the rats were euthanized, and the calvaria with implanted specimens were harvested.

Specimens were dehydrated in a graded concentration of ethanol and embedded in methyl methacrylate resin (Technovit^®^ 7200; Heraeus Kulzer, Wehrheim, Germany). Blocks of methyl methacrylate, including the samples, were sectioned at 100 μm thickness in the sample’s long axis using a diamond-coated saw cutter and the Exakt grinding system (EXAKT Advanced Technologies GmbH, Norderstedt, Germany). The sections were further ground and polished to 20 μm thickness. The sections were then stained with H&E and basic fuchsin, and the bone was stained red. 

H&E staining and immunofluorescence staining of periostin, HLA class I, vWF, and CEMP1 were performed. For the analyses, enucleated calvaria were fixed with 4% paraformaldehyde solution and decalcified using 10% sodium citrate and 22.5% formic acid for 2 weeks at 4 °C. Staining was performed on 6 μm paraffin-embedded sections.

After deparaffinization, the slides were incubated with Proteinase K (10 μg/mL, AM2546, Thermo Scientific, Waltham, MA, USA) for 20 min at 37 °C or for GFP, with pepsin (Digest-All™ 00–3009, Invitrogen, Waltham, MA, USA) for 10 min at 37 °C. Subsequently, the slides were incubated with antibodies against periostin (1:1000 diluted, ab14041, Abcam, Cambridge, MA, USA), vWF (1:100 diluted, AB7356, EMD Millipore Co., Waltham, MA, USA), HLA (1:500 diluted, ab70328, Abcam plc, UK), and CEMP1 (1:1000 diluted, sc-53947, Santa Cruz Biotechnology, Berkeley, CA, USA) at 4 °C overnight. The specimens were sequentially incubated with secondary antibodies and streptavidin peroxidase. The results were visualized following staining with a diaminobenzidine (DAB) reagent kit (Invitrogen, Waltham, MA, USA). The sections were counterstained with Mayer’s hematoxylin. Negative control staining was performed without primary antibody. All specimens were observed using a stereomicroscope (MD5500D; Leica, camera: DFC495; Leica, Lens: HCX PL APO 409; Leica). The total length of each sample (TL) was measured at undecalcified slide and the length of the portion in contact with the periodontal ligament like soft tissue (SL) and bone (BL) was measured to analyze the ratio of the portion in contact with the bone.

### 2.9. Statistical Analysis

Statistical analysis was performed by one-way analysis of variance (ANOVA) followed by Bonferroni’s multiple comparison test using SPSS (ver.18) software for a comparison between the groups. *p* < 0.05 was used as the significance level.

Difference of length of the portion in contact with the periodontal ligament like soft tissue between the seeding group (G1, G2) and the printing group (G3, G4) was compared using Kruskal–Wallis analysis followed by Jonckheere–Terpstra test using SPSS (ver.18) software for a comparison between the groups. *p* < 0.05 was used as the significance level. Description of each group were summarized in Table 1. 

## 3. Results

### 3.1. Characterization of hPDLSCs

The differentiation of hPDLSCs was evaluated at 3 weeks (Figure 1A–C). Flow cytometric analysis showed that approximately 37.49% of the hPDLSCs expressed CD34, 92.6% expressed CD13, 94.52% expressed CD90, and 80.71% expressed CD146 (Figure 1D). CD34 is known as an MSC^−^ marker, which marks primitive endothelial cells and hematopoietic progenitors. On day 7 of culture, the expression of CEMP1 in the cell printing groups (G3, G4) was significantly higher than that in the cell seeding groups (G1, G2) (Figure 1E). Neither COL1 nor ALP showed any differences. 

### 3.2. In Vitro Experiments

Bioink, a blend of hPDLSCs and 4% collagen printed on a 3D-printed titanium scaffold, was cultured for 21 d under gelation conditions. The gelled bio-ink did not collapse but remained in its originally-printed form in all groups until 21 d (Figure 2A–D). SEM showed that in the seeding groups (G1, G2), PDL cells had no direction and were not well organized, but in the printing groups (G3, G4), PDL cells were well aligned and had direction (Figure 2E–H). Live/dead cell assay showed that in cell seeding groups (G1, G2), the cell distribution was uneven, and cell aggregation was observed. In the cell printing groups (G3, G4), the cell distribution was homogeneous and was confirmed to consist of single cells without cell aggregation (Figure 3A–H). As a result, we confirmed that the cell printing method was more reliable than seeding. In the CCK-8 assay, shear stress occurred during printing and thus cell viability was lower than that in the cell seeding group (G1, G2). However, in the cell printing groups (G3, G4), proliferation occurred to a good extent on day 7 (Figure 3I).

### 3.3. PDL Regeneration In Vivo

To verify the formation of periodontal ligament-like connective tissues on titanium implant from human periodontal stem cells in vivo, we transplanted the 3D printed porous scaffold with bioprinted human PDL-derived cell into the calvaria of athymic rat femur for 6 weeks. All the seeding group (G1, G2) and the printing group (G3, G4) titanium implants with periodontal ligament were successfully stabilized on the calvaria of the athymic rat. 

At 6 weeks after transplantation, the 3D-printed titanium scaffolds were covered with fibrous connective tissue between the rat calvaria bone and 3D-printed titanium scaffold in the cell printing groups (G3, G4) (Figure 4). Fibrous connective tissue was not observed in the seeding groups (G1, G2) (Figure 4). On the contrary, fibrous connective tissue was apparent in the cell printing groups (G3, G4) (Figure 4). PDL-like tissue was oriented parallel to the porous titanium implant surface in group 3 and 4. Osseointegration predominated on the titanium surface in group 1 and 2 (Figure 4). With H&E and basic fuchsin staining, new bone formation into the porous scaffold was evident in the seeding groups; however, in the printing groups, fibrous connective tissue was evidently visible between the host bone and the scaffold (Figure 4 and Figure 5). The cellular morphology in group 3 and 4 was analyzed by hematoxylin and eosin (H&E) staining. Periodntal ligament-like cells were uniformly distributed adjacent to the calvarial bone (Figure 5). Immunohistochemistry revealed that HLA, periostin, vWF, and CEMP1 were expressed in the connective tissues produced in the cell printing groups (G3, G4) (Figure 6). However, the positive cells varied in number and distribution pattern depending on the target molecules. Since PLD-like fibrous connective tissue was not observed in the seeding groups (G1, G2), Immunohistochemistry stain was not performed in seeding groups. 

For HLA, a moderate positive stain was observed adjacent to the calvaria in Group 3 and 4. For periostin, a relatively strong positive reaction was observed adjacent to the bone. A small number of vWF-positive cells were detected in the inner cells of tge periodontal ligament-like cellular strands. Cemp 1 positive cells were also confirmed in the innermost cells (Figure 6). According to this study, FGF2 did not play a role in the regeneration of periodontal ligaments with bioprinting. 

New bone was observed on the titanium implant surfaces in the seeding group (G1, G2). The total length of each sample (TL) was measured at the undecalcified slide, and the length of the portion in contact with the periodontal ligament-like soft tissue (SL) and bone (BL) was measured to analyze the ratio of the portion in contact with the bone (Figure 7). The SL/TL of the seeding group (G1, G2) was 9.4%, and in the printing group (G3, G4) was 91.2%. Statistically, significantly more periodontal ligament like soft tissue-implant interfaces were formed in the printing group than in the seeding group (*p* < 0.05).

## 4. Discussion

Organs function through cooperation with adjacent tissues and other organs [18]. Fibrous connective tissues are essential in achieving organ functions, including tight connectivity, mobility, and resistance against mechanical stimuli and infections [19]. When reconstructing a mandibular defect with a titanium 3D-printed implant, there is a surface that contacts the implant and the existing bone. Reconstruction with 3D-printed hybrid artificial organs can be performed to prevent infection if the fibrous connective tissue, such as the periodontal ligament, is interposed between the titanium implant and the bone in the area close to the oral mucosa, and osseointegration is obtained at the area contacting the basal bone at the lower part. 3D-printed hybrid artificial organs can also be applied to dental implants. 3D-printed hybrid dental implants could establish physiological tooth functions, including the ability to react to mechanical stimuli and the ability to resist infection. 

No studies have been attempted to regenerate PDL by printing PDL cells on the surface of titanium implants. Currently, 3D cell-printing techniques have been utilized as new biofabrication platforms because of their ability to precisely locate living cells in pre-defined spatial locations with scaffolds and various growth factors [10]. 3D bioprinting is the utilization of 3D printing and 3D printing–like techniques to combine cells, growth factors, and biomaterials to fabricate biomedical parts that maximally imitate natural tissue characteristics [10]. A 3D bioprinted skin patch mixed with adipose-derived stem cells enhanced wound closure, re-epithelization, and neovascularization [10]. Implanted, bioprinted, neural stem cells in adult zebra fish accelerated repair of traumatic brain injury and restoration of function [11]. Implanted, bioprinted cell types including human amniotic-derived stem cells, rabbit ear chondrocytes, and rabbit myoblasts produced newly formed vascularized bone tissue, cartilage, and muscle with physiologically relevant mechanical properties [12].

Various experiments were carried out to confirm that bioprinting of periodontal ligaments is an effective method for regenerating PDL on 3D printing titanium. SEM showed that in the seeding groups (G1, G2), PDL cells had no direction and were not well organized, but in the printing groups (G3, G4), PDL cells were well aligned and had direction. Live/dead cell assay demonstrated that in cell seeding groups (G1, G2), the cell distribution was uneven and cell aggregation was observed. In the cell printing groups (G3, G4), the cell distribution was homogeneous and confirmed to consist of single cells without cell aggregation (Figure 2).

PDL-derived cells possess the ability of periodontal regeneration, including cementum formation [13]. CEMP-1 was known as one of the cementum marker genes found to be expressed in cementoblasts, PDL cells, and cells around vascular networks [20,21]. On day 7 of culture, the expression of CEMP1 in the cell printing groups (G3, G4) was significantly higher than in the cell seeding groups (G1, G2) (Figure 1). This finding suggests that the printed PDL cells have the capacity for cementogenesis induction. At 6 weeks after transplantation, the 3D-printed titanium scaffolds were covered with connective tissue between the rat calvaria and scaffold in the cell printing groups (G3, G4) (Figure 3). In decalcified tissue specimens, organized connective tissue was not observed in the seeding groups (G1, G2) (Figure 2). In contrast, well-organized connective tissue was apparent in the cell printing groups (G3, G4) (Figure 4). In undecalcified tissue specimens with H&E staining and basic fuchsin staining, new bone formation into the porous scaffold was evident in the seeding groups, but in the printing groups, well-organized connective tissue was obvious between the rat calvaria and scaffold (Figure 3). Based on the results of this study, we conclude that PDL bioprinting technology is a reliable method for the regeneration of PDL on titanium 3D-printed scaffolds. Because the athymic rat model is versatile for the validity assessment of human cell regeneration, we chose it to assess periodontal tissue regeneration induced by human PDL cell printing on 3D-printed titanium scaffolds. It was also necessary to confirm that newly formed PDL fibers are interconnected between the titanium and the bone, with a structure similar to that seen with natural periodontal tissue and the rat calvaria is appropriate for the observation of PDL formation because of its profound blood supply. Immunohistochemical staining revealed that HLA, periostin, vWF, and CEMP1 were expressed in the connective tissues produced in the cell printing groups (G3, G4) (Figure 6). Taken together, these results suggest that newly produced connective tissue between 3D-printed implants and calvaria bone has PDL characteristics and originated from hPDLSCs.

## 5. Conclusions

Cell printing technology, rather than seeding PDL cells, has produced a periostin-positive-connective tissue interface between 3D-printed titanium scaffold and the bone. Reconstruction with 3D-printed hybrid artificial organ using bioprinting can be performed to prevent infection if the fibrous connective tissue, such as the PDL, is interposed between the titanium implant and the bone in the area close to the oral mucosa. This study shows the potential for a next-generation bio-implant coated with printed periodontal ligament resembling a natural tooth using titanium 3D printing and PDL bioprinting for treating tooth loss.

## Figures and Tables

**Figure 1 cells-10-01337-f001:**
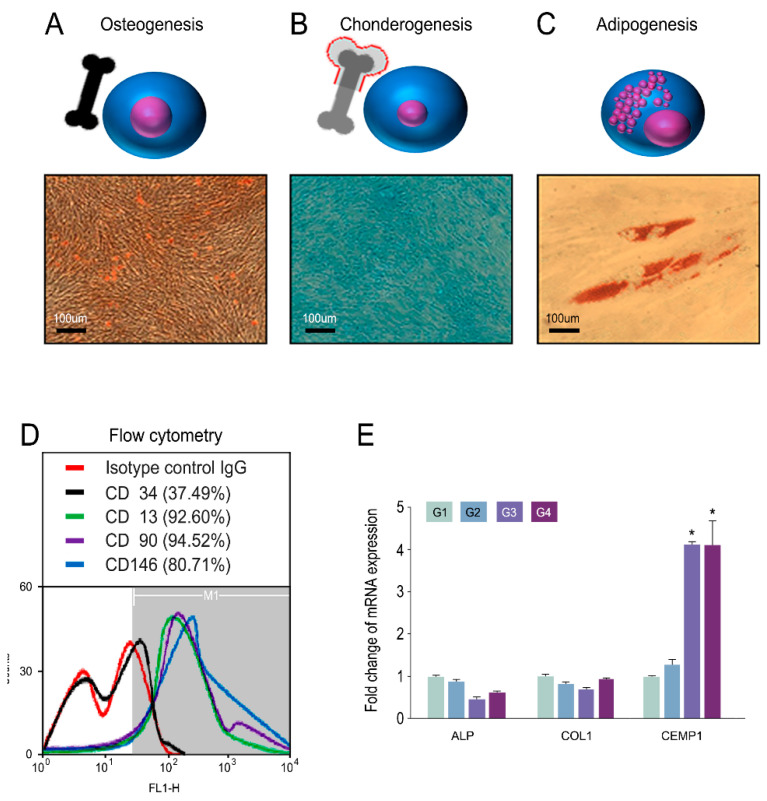
Cell characterization of hPDLSCs. (**A**–**C**) Tripotent lineage of hPDLSCs. (**A**) Alizarin red S staining for osteogenesis; (**B**) Alcian blue staining for chondrogenesis; (**C**) Oil red O staining for adipogenesis; (**D**) flow cytometry using mesenchymal stem cell markers CD34, CD13, CD90, and CD146; (**E**) quantitative PCR analysis of hPDLSCs in titanium scaffold at 7 d after cell seeding or bioprinting. * *p* < 0.05.

**Figure 2 cells-10-01337-f002:**
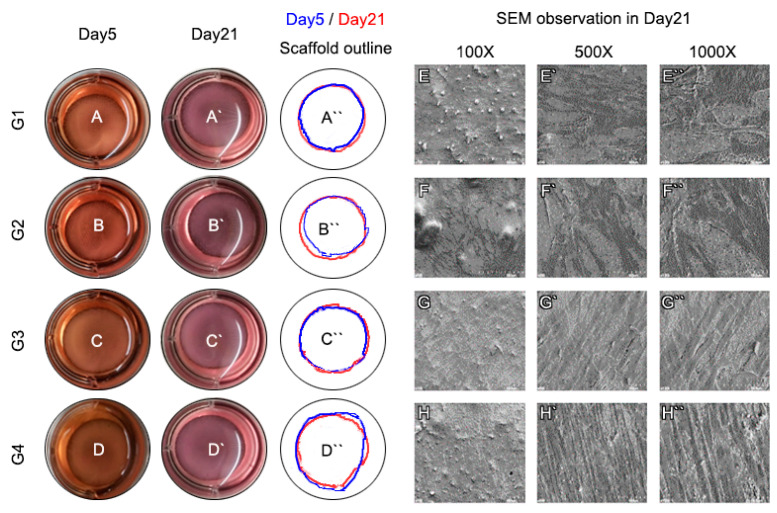
Sample observation. (A–D) The gelled bioink did not collapse but retained its original printed form in all groups till 21 d. (E–H) SEM showed that in the seeding groups (E,F), PDL cells had no directional alignment and were not well organized. However, in bioprinting groups (G,H), PDL cells were well aligned and had directional alignment.

**Figure 3 cells-10-01337-f003:**
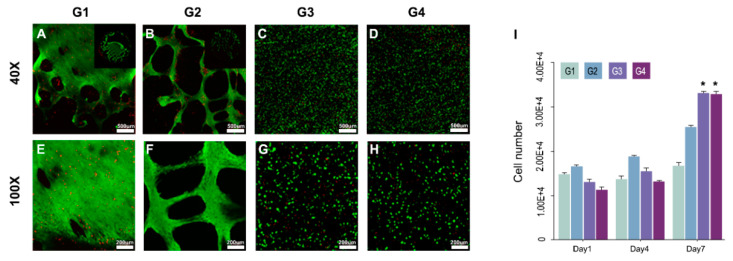
Cell viability and proliferation test. Live and dead assay results of cell-printed structure at day 1, magnification 100×, green dot: live cell, red dot: dead cell (scale bar: 500 μm and 200 μm). (**A**,**B**,**E**,**F**) cell seeding group showed irregular distribution; (**C**,**D**,**G**,**H**) bioprinting group showed well distribution of cells.; (**I**) In the CCK-8 assay, shear stress occurred during printing, and cell viability was lower than that of the cell seeding group on day 1 but showed no significant difference (G1, G2). However, proliferation of cell printing group (G3 and G4) has occurred to a good extent on day 7. * *p* < 0.05.

**Figure 4 cells-10-01337-f004:**
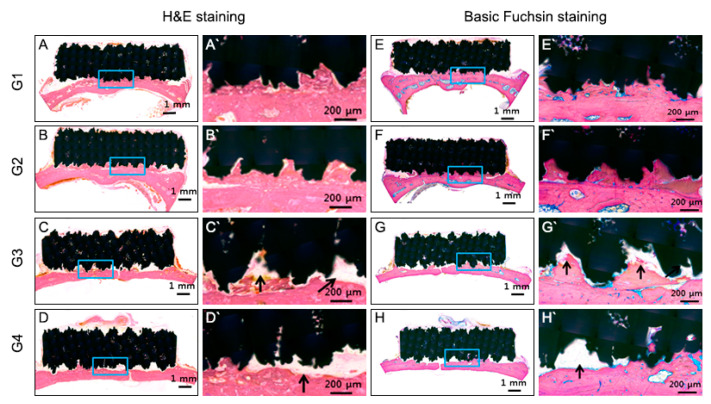
Histological analysis of undecalcified in vivo sample. (**A**–**D**) H&E staining. (**E**–**H**) Basic Fuchsin staining. New bone formation into porous scaffold was evident in (**A**,**B**) and (**E**,**F**) but not in (**C**,**D**) and (**G**,**H**) well organized fibrous connective tissue was obvious between rat calvaria and scaffold (black arrow **C**,**D**,**G**,**H**).

**Figure 5 cells-10-01337-f005:**
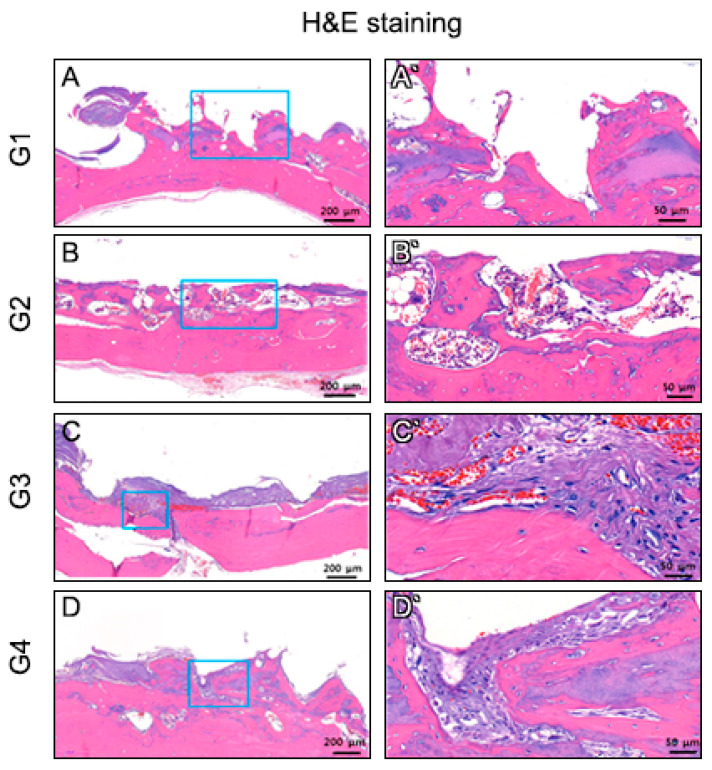
H&E staining of decalcified cross-section samples. In decalcified tissue specimen, fibrous connective tissue was not observed in the seeding group (**A**,**B**). On the contrary, well-organized fibrous connective tissue was apparent in the cell printing group (**C**,**D**).

**Figure 6 cells-10-01337-f006:**
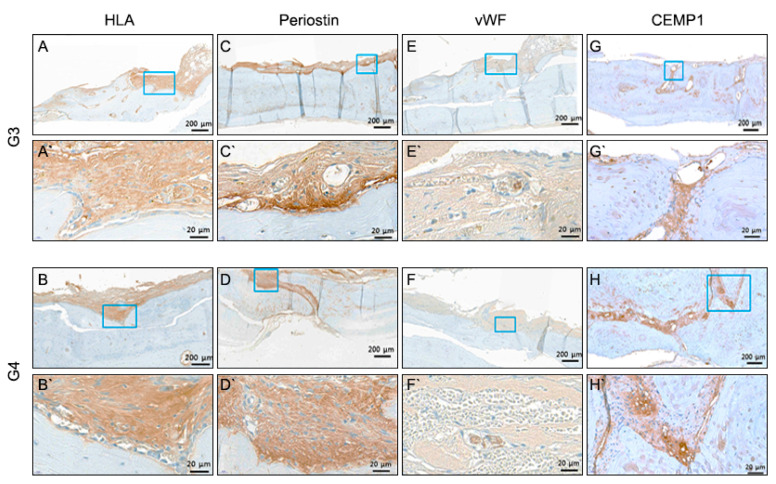
Immunohistochemistry of bioprinting group. (**A**,**B**) HLA specific staining; (**C**,**D**) Periostatin specific staining; (**E**,**F**) vWF specific staining; (**G**,**H**) CEMP1 specific protein. All four markers show the presence of connective tissue, which is evidence of periodontal ligament regeneration.

**Figure 7 cells-10-01337-f007:**
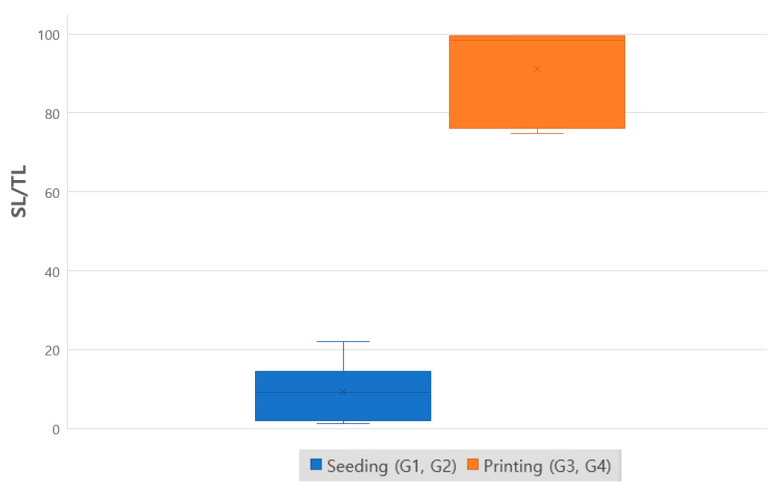
Comparison of SL/TL. The ratio of the total length of each sample (TL) and length of the portion in contact with the periodontal ligament-like soft tissue (SL) by histological analysis of undecalcified in vivo sample. (*p* < 0.05).

**Table 1 cells-10-01337-t001:** Description of each group.

Group 1	Group 2	Group 3	Group 4
Titanium scaffold + collagen printing + cell seeding	Titanium scaffold + collagen printing + FGF 2 + cell seeding	Titanium scaffold + Cell printing	Titanium scaffold + FGF 2 + Cell printing

## Data Availability

All results generated or analyzed during the present study are included in this published article. Data and materials will be made available upon request via email to first author (davidjoy76@gmail.com).

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
