# Peer review of "Bioprinting on 3D Printed Titanium Scaffolds for Periodontal Ligament Regeneration"

_cells, 2021, doi:10.3390/cells10061337_

Round 1
Reviewer 1 Report
In this work, the authors propose to investigate periodontal ligament regeneration on the surface of titanium using human periodontal ligament stem cells (hPDLSCs) and there-dimensional (3D) cell-printing technique. They presented supporting evidence demonstrating that 3D printed scaffold with bioprinted hPDLSCs can fabricate PDL-like fibrous connective tissue and may also have ability cementogenesis. These results suggest that 3D-printed hybrid dental implants may be similar in structure to healthy periodontal tissue and be new strategy for prosthodontic treatment for missing teeth. However, some major issues need to be coped with to allow publication.
Major comment:
- Because it is hard to understand each procedure and group in in vivo study, the authors had better make a schema for the detail how you transplanted the cells and materials.
- In the materials and methods, although the authors state that “Non-decayed human third molars that had been impacted in the mandible were extracted from 5 adults (18–28 years of age)…”, which cells did you use for experiments? Did you use all or choose specific one vial? If you choose one vial for in vivo experiments, how did you decide it? Please clear these points. Furthermore, did you test the differentiation and proliferation capacity of all PDLSCs? Can you please show quantitative values for all donors or at least representative images for all PDLSCs cultures?
- 1 – If the cells used in this study are stem cells (mesenchymal stem cells: MSCs), it is important that the cells meet at least the minimum criteria of the ISCT, i.e:
1.) Adhesion to plastic - this is fine and proven in the manuscript.
2.) Specific surface antigen (Ag) expression - the ITSC suggests flow cytometry here, please show the corresponding data. For example, authors should investigate the MSCs positive markers not only CD 90 but also CD 73, 105. Furthermore, MSCs usually lack expression of CD34. However, 37.49% of the cells used in this study was positive for CD34. Why are there many positive cells for CD34?
3.) In vitro differentiation: this is fine and proven in the manuscript.
- 4 and 5 - In the discussion the authors state that “On day 7 of culture, the expression of CEMP1 in the cell printing groups (G3, G4) was significantly higher than in the cell seeding groups (G1, G2) (Figure. 1). This finding suggests that the printed PDL cells have the capacity for cementogenesis induction.”. Why did not you investigate newly cementum-like tissue on the titanium scaffold surface in histological analysis?
- 4, 5, and 6 – The authors should quantitate histological data of each group by measuring staining area or counting specific cells. And then, the authors should also mention about number of rats used in each experiment and perform statistical analysis among each group.
Minor comments:
- The authors should mention about all procedure of experiments and materials used in this study in Materials and Methods section. (i.e: te detail of differentiation assay and flow cytometric analysis for PDLSCs)
- Page 2, line 90 – Correct: “2.2. 3D printing”
- 3 A-H – How many days did these cells incubate? The authors should mention about that in Results section or figure legends. Furthermore, Fig. 3 A-H were not mentioned in Results section.
- Page 7, line 221 – “Col1” should be “COL1”.
- Page 8, line 258 – Since hematoxylin and eosin staining has already been mentioned before this line, it should be changed to “H&E staining”.
- Page 10, line 322 – Correct: … groups (G3, G4) (Figure 6). Taken together,…
Author Response
Thank you very much for the review of the manuscript titled above. We have now carried out substantial revision according to the reviewer’ comments, which the details of the corrections are summarized below.
We are very grateful for the review of the manuscript and comments that improved the quality of the manuscript, and hope that the corrections would be adequate for this articles to be accepted by your journal.

Reviewer 2 Report
see attached file

Author Response

(The authors gave the same response as above.)

Round 2
Reviewer 1 Report
Thank you for the revised version of your manuscript.
Comment 1:
It is hard to find newly cementum-like tissue on the titanium scaffold surface in your new data. If you don't have any representative data/image for proving it, you should not mention about it.
Comment 2:
page 2 line 99; "1×106 cells" -> "1×106 cells"
I accepted the other answers to my comments.
Author Response
Thank you for providing us the opportunity to submit a revised version of our manuscript. Below, we have addressed each point raised by the reviewers and described the changes made to the manuscript.

Reviewer 2 Report
I'm satisfied with authors' answers and the new revision of the text. Thank you.
Author Response
We appreciate your time and valuable advice to improve manuscript in round1. Thank you very much.
